# Study of Modeling and Optimal Take-Off Scheme for a Novel Tilt-Rotor UAV

**DOI:** 10.3390/s22249736

**Published:** 2022-12-12

**Authors:** Zelong Yu, Jingjuan Zhang, Xueyun Wang

**Affiliations:** 1School of Instrumentation Science and Opto-Electronics Engineering, Beihang University, Beijing 100191, China; 2School of Aeronautic Science and Engineering, Beihang University, Beijing 100191, China

**Keywords:** blended wing body, optimal take-off scheme, tilt-rotor, trajectory planning

## Abstract

The optimal trajectory planning for a novel tilt-rotor unmanned aerial vehicle (UAV) in different take-off schemes was studied. A novel tilt-rotor UAV that possesses characteristics of both tilt-rotors and a blended wing body is introduced. The aerodynamic modeling of the rotor based on blade element momentum theory (BEMT) is established. An analytical method for determining the taking-off envelope of tilt angle versus airspeed is presented. A novel takeoff–tilting scheme, namely tilting take-off (TTO), is developed, and its optimal trajectory is designed based on the direct collocation method. Parameters such as the rotor thrust, tilt angle of rotor and angle of attack are chosen as control variables, and the forward velocity, vertical velocity and altitude are selected as state variables. The time and the energy consumption are considered in the performance optimization indexes. The optimal trajectories of the TTO scheme and other conventional schemes including vertical take-off (VTO) and short take-off (STO) are compared and analyzed. Simulation results indicate that the TTO scheme consumes 47 percent less time and 75 percent less energy than the VTO scheme. Moreover, with minor differences in time and energy consumption compared to the STO scheme, but without the need for sliding distance, TTO is the optimal take-off scheme to satisfy the flight constraints of a novel tilt-rotor UAV.

## 1. Introduction

A tilt-rotor aircraft combines the vertical take-off and landing capabilities of helicopters with the high-speed cruising capabilities of fixed-wing aircraft by tilting the motors. This relatively large flight envelope allows them to be exploited in a wide range of missions with various objectives, making them popular in both civilian and military fields [1]. However, structural design principles such as vortex mode stability and drive mechanisms that must have sufficient inner wing volume can result in very stiff, short-span wings and thick airfoils [2].

Tilt-rotor UAVs can be categorized by rotor type, wing type, body type and ducted fan type [2]. The tilt-rotor and tilt-wing designs are the two most common structures. On the one hand, both can maintain flight stability during the transition from hovering to forward flight, but on the other hand, the most important differences between them are aerodynamic drag and the required power of the rotor in the take-off and transition process [3,4,5]. A type of tilt-rotor aircraft, the V-22 Osprey is currently in service in the military as a practical aircraft due to its advantages [4]. The disadvantage of the tilt-rotor design is that the wing becomes a body with a large drag coefficient as the hover height changes, resulting in more energy consumption by the UAV to gain altitude [6]. Muraoka K et al. also designed and flight-tested a typical tilt-wing UAV [7]. The wings of the UAV tilt with the rotors, minimizing the lift loss because of the downward slipstream in hover mode, which is also beneficial during the rotor-tilting process. However, the moment of inertia of the entire wing is larger than that of the rotor, assuming that both the wing and the rotor are rigid bodies and that their center-of-mass positions coincide. Thus, according to the theory of rigid body rotation $W=J\cdot \omega^{2}/2$, where W is work of moment, J is inertia, and ω is tilting angular velocity, the moment of the wing does more work than the moment of the rotor when tilting at the same angle with the same angular velocity. Both are powered by a battery that tilts the servo motor; therefore, we believe that tilting the entire wing requires more energy than tilting only the rotors. Especially for all-electric UAVs, the battery is the primary energy source. Limited by the weight and load of the battery, it is necessary to minimize the energy consumption by the actuators other than the power system. Therefore, the tilt-wing UAV’s disadvantage is that tilting wing motors consume more energy, resulting in reduced overall endurance in all flight processes [8].

A novel tilt-rotor UAV was designed to overcome the above problems. The work presented in this paper focuses on the novel tilt-rotor UAV depicted in Figure 1, which combines a blended wing body (BWB) and tilt-rotor (TR). This UAV, namely the blended wing body tilt-rotor (BWTR), is equipped with three rotor tilting mechanisms, which are mounted at the front and back of the fuselage. In addition, one part of the wing for the BWTR is fixed to the body at a neutral position for the optimum forward flight efficiency, and the other part of the wing tilts with the rotor. This design minimizes the aerodynamic drag and increases the endurance of the UAV due to tilting part of the wing together instead of tilting the whole wing.

In addition, the takeoff–tilting transition process is very important for all tilt-rotor UAVs. At present, research on the takeoff–tilting transition process is mainly focused on the methods of design, modeling, and control [8,9,10,11]. There are few investigations on the flight performance comparative analysis of different takeoff- tilting schemes and trajectories. An optimal takeoff–tilting scheme and trajectory can not only solve the problem of control redundancy, but also improve the stability of body attitude and help reduce the time and energy consumption [12].

At present, there are two main take-off tilting schemes: vertical take-off VTO) and short take-off (STO) [12,13,14]. For the VTO scheme, its advantages are the low requirement for takeoff site and the simple control of the tilting rotors at a fixed altitude. However, the VTO scheme also results in a higher power requirement for the UAV to climb. The payload of a UAV is limited due to the maximum engine power [15]. Compared with the VTO scheme, the STO scheme makes full use of the lift force to improve the payload of the UAV. At the same time, the UAV has higher stability because of the forward flying speed. However, it needs a certain sliding distance and, therefore, requires the runway [16].

By analyzing the advantages and disadvantages of the two schemes, a novel takeoff–tilting scheme i.e., tilting take-off (TTO), was developed for the BWTR in this paper. Moreover, optimal trajectory planning is an efficient way to improve the flying performance of an aircraft [17,18,19]. The optimal trajectory planning problem of the tilt-rotor UAV in the takeoff–tilting transition process can be described as follows: find an optimal flight trajectory among all the takeoff–tilting transition flight trajectories that meets the flight requirements and minimizes the cost (performance indexes) [20,21,22,23]. Therefore, the optimal takeoff–tilting scheme and trajectory of the BWTR can be obtained by comparing the optimal trajectory planning results for different takeoff–tilting schemes under the same constraints [24,25].

In this paper, the modeling and takeoff optimal trajectory planning in different take-off schemes were researched for BWTR. The main contributions of the paper include:Firstly, aiming at the shortcomings of tilt-rotor UAV and tilt-wing UAV, a novel tilt-rotor aircraft BWTR is introduced and its dynamics are modeled. The effect of axial air flow velocity is considered to establish a more accurate rotor mathematical model. In addition, an analytical method to determine the tilt angle-airspeed envelope of tilt-rotor UAV is presented.Second, we investigate the VTO scheme and STO scheme for the novel UAV. Aiming at improving the two schemes, a novel takeoff–tilting scheme, i.e., tilting take-off (TTO), is proposed and researched.Finally, optimal trajectory planning is carried out for the three schemes by using the direct collocation method. The optimal scheme and trajectory in the takeoff–tilting process are obtained in simulations, and comparative analysis of the optimization results are presented.

## 2. Mathematical Modeling

Figure 2 illustrates the structure of the mathematical modeling framework we investigated. Mathematical modeling, which improves the efficiency of solving optimal trajectory planning problems, consists of four main components: description of the model, definition of the coordinate system, dynamics model, and the tilt angle-velocity envelope model.

A brief description of each component is given below. The structure, layout and actuators of the novel tilt-rotor UAV are introduced, and the model parameters of the body and actuators are described in the description of the model. The definition of the coordinate system describes all of the coordinate frames adopted by the dynamic model and tilt angle-airspeed envelope model. Then, the dynamic model includes kinematic equations and actuator models, with the rotor model and aerodynamic model being the main actuator models. A tilt angle-airspeed envelope model is developed to obtain the tilt corridor of the UAV using force and moment balance analysis.

The description of the model and definition of the coordinate system are the basis for the dynamic model and tilt angle-airspeed envelope model. Depending on the model assumptions and the trajectory characteristics of the different schemes, the dynamic model can be simplified to equations of state of the trajectory optimization planning. Moreover, energy consumption is an essential component of performance parameters for trajectory optimization planning. We can calculate the energy consumption by using the required power obtained by the rotor model. In addition, the constraints for optimal trajectory planning are determined by the tilting corridor of the UAV and rotor model.

### 2.1. Description of Model

As shown in Figure 3, the structure of the BWTR is longitudinally symmetric, and its takeoff–tilting process is completed in the longitudinal plane. Differing from traditional UAVs, the BWTR adopts a blended wing body configuration, which improves the lift-drag ratio and stealth performance. Besides this, the normal horizontal rear elevator and rudder are eliminated, which means fewer mechanisms, lower cost, more energy savings, longer flight durations and higher control requirements. The two main rotors fixed to the aircraft frame in the forward part rotate in opposite directions, decreasing the reaction torque generated to almost zero. Moreover, they can tilt from the vertical position ($A_{rl}=0^{\circ }$) to horizontal position ($A_{rl}=90^{\circ }$) or inversely through a tilting mechanism. Unlike the general UAVs, the new configuration adopts a smaller rear rotor to meet the low thrust-weight ratio requirement during the FW phase, which reduces the UAV weight and improves the cruising power efficiency. However, to ensure sufficient thrust during the VTOL phase, the rear rotor should have a longer moment arm, which poses greater challenges for control precision.

According to the above characteristics of the BWTR, the mathematical modeling of the UAV is studied in this paper.

The model parameters of the tilt-rotor UAV are specified in Table 1.

### 2.2. Definition of Coordinate System

Figure 2 is the schematic of the BWTR. The coordinate systems are defined as follows: $O_{b}=X_{b}Y_{b}Z_{b}$ is the body-fixed coordinate frame (BFF). $O_{b}$, is at the center of gravity, $X_{b}$ pointing to the right of the aircraft, $Y_{b}$ pointing front and $Z_{b}$ pointing down. $O_{n}=X_{n}Y_{n}Z_{n}$ is the navigation coordinate frame, which coincides with the North-East-Down geographic coordinate frame (NED). $O_{mr}=X_{mr}Y_{mr}Z_{mr}$,$O_{ml}=X_{ml}Y_{ml}Z_{ml}$,$O_{mb}=X_{mb}Y_{mb}Z_{mb}$ are the motor coordinate frames (MCF) which are fixed with three rotors, respectively. $O_{w}=X_{w}Y_{w}Z_{w}$ is the wind axis coordinate frame (WAF). $O_{b}$ is the original point of the tilt-rotor UAV, and the coordinate is ${\left(0,0,0\right)}^{T}$. Therefore, the coordinates of the right rotor are ${\left(x_{r},y_{r},z_{r}\right)}^{T}$, of the left one ${\left(x_{l},y_{l},z_{l}\right)}^{T}$, and of the rear one ${\left(x_{b},y_{b},z_{b}\right)}^{T}$.

### 2.3. Dynamic Model

In this paper, the following assumptions need to be made prior to dynamics modeling:

Since the novel tilt-rotor UAV is longitudinally antisymmetric, it can be assumed that: if the UAV is symmetric about the $X_{b}$-$Z_{b}$ plane of the body coordinate system, then the inertia products $I_{xy}$ and $I_{zy}$ of the UAV are both zero.The rotor system consists of rotor, motor and tilting mechanism, with the center of gravity balanced on the tilt axis. As a result, the center of gravity position of the rotor system remains practically unchanged when the rotor tilt angle varies from 0 to 90 degrees. In addition, the mass of the rotor system accounts for about five percent of the weight of the entire aircraft, which further reduces the effect of the rotor tilting relative to the center of gravity position of the UAV. Hence, it can be assumed that: the change in the position of the center of gravity of the UAV caused by the tilting of the motor can be ignored.

According to the above model assumptions, utilizing the standard Newton–Euler formulation, the equations of dynamics for the UAV are as follows.
(1)$$\begin{array}{l} {\vec{F}}^{b}=\frac{d}{dt}\left(m{\vec{V}}^{b}\right)=m\left(\frac{\delta {\vec{V}}^{b}}{\delta t}+{\vec{\Omega }}^{b}\times {\vec{V}}^{b}\right)\\ {\vec{M}}^{b}=\vec{I}\left(\frac{\delta {\vec{\Omega }}^{b}}{\delta t}\right)+{\vec{\Omega }}^{b}\times \left(\vec{I}{\vec{\Omega }}^{b}\right)\\ {\vec{\dot{X}}}^{n}=C_{n}^{b}{\vec{V}}^{b}\\ \vec{\dot{\Theta }}=J_{n}^{b}{\vec{\Omega }}^{b} \end{array}$$
where ${\vec{F}}^{b}$ is the BFF-based total force vector, ${\vec{M}}^{b}$ is the BFF-based total moment vector, $\vec{V}={\left(u,\;v,\;w\right)}^{T}$ is the BFF-based velocity vector, $\;\vec{\Omega }={\left(p,\;q,\;r\right)}^{T}$ is the BFF-based angular rotation rate vector, ${\vec{X}}^{n}={\left(x^{n},\;y^{n},\;z^{n}\right)}^{T}$ is the NED-based position vector, and $\vec{\Theta }={\left(\phi ,\;\theta ,\;\psi \right)}^{T}$ the is Tait–Bryan rotational angle vector. Besides, m is the total mass and $\vec{I}$ is the moment of inertia matrix. Furthermore, $C_{n}^{b}$ is the BFF-NED translational velocities transformation matrix and $J_{n}^{b}$ is the Tait–Bryan rotational angle rate transformation matrix.

#### 2.3.1. Rotor Model

The rotor is the main power source of the tilt-rotor UAV. In the whole flight process, the rotor is a lift mechanism, a maneuvering mechanism, and a thrust mechanism. In the optimal trajectory planning of the tilt-rotor UAV, the rotor thrust is the main control variable, and the rotor power is one of the important parameters of the cost function.

The static thrust of the rotors can be measured by a thrust test stand. However, when the UAV has airspeed, the axial air flow velocity of the rotors will affect the propeller’s angle of attack, which in turn will affect the thrust. That is, when the speed of motor is certain, the thrust of the rotors will also vary with the airspeed of the UAV. In this section, the rotors are modeled by using the blade-element theory.

Figure 4 shows the force analysis diagram of the blade-element theory. Generally, the hub portion of the propeller does not generate thrust and accounts for about 20% of the overall propeller diameter. The blade primitive is selected as a profile $r_{p}$ away from the propeller’s tip, and its force analysis can be analogous to that of a wing. $V_{a}$ is rise velocity of the blade element, $W_{a}$ is relative velocity of the airflow, $\beta_{p}$ is propeller blade angle, $\phi_{p}$ is airflow angle, $\alpha_{p}$ is propeller blade angle of attack, $\omega_{p}$ is angular velocity of propeller rotation. Table 2 summarizes all of the main parameters used in the rotor modeling.

The lift and drag of the rotor blades are mainly related to the airfoil shape and rotation speed of the propeller blades, which can be denoted as:(2)$$\begin{array}{l} C_{LP}=\frac{K_{0}\alpha_{P}}{1+K_{0}/\pi A}\\ C_{DP}=c_{fd}+\frac{1}{\pi Ae}C_{L}{}^{2}\\ L_{p}=\frac{1}{2}\rho V_{a}^{2}C_{L}S_{sa}\\ D_{p}=\frac{1}{2}\rho V_{a}^{2}C_{D}S_{sa} \end{array}$$
where ρ is the air density, $C_{LP}$ is the lift coefficient, $C_{DP}$ is the drag coefficient, $S_{sa}=\lambda B_{p}r_{p}c_{p}$ is the blade area, $A=2r_{p}/c_{p}$ is the spreading ratio; the constant $K_{0}\approx 6.11$, and the Oswald factor e takes values in the range 0.7~0.9.

According to the blade-element theory shown in Figure 4, the total rotor thrust and power requirement are given as follows:(3)$$\begin{array}{l} T=\int dL_{p}\cos\phi_{p}-\int dD_{p}\sin\phi_{p}\\ P=\left(\int dL_{p}\sin\phi_{p}+\int dD_{p}\cos\phi_{p}\right)r\omega_{p} \end{array}$$

The thrust of the rotors, which can be calculated by Equations (2) and (3), can be defined as $T_{r}{}^{m}={\left(0,\;0,\;T_{r}\right)}^{T}$, $T_{l}{}^{m}={\left(0,\;0,\;T_{l}\right)}^{T}$ and $T_{b}{}^{m}={\left(0,\;0,\;T_{b}\right)}^{T}$ in the MCF. As well, taking the right rotor as an example, according to the definition of coordinate systems, the thrust vector of the right rotor is translated from the MCF to BBF, and the moment is further calculated under the machine system.
(4)$$\left\{ \begin{array}{l} \left[\begin{array}{c} T_{xr}{}^{b}\\ T_{yr}{}^{b}\\ T_{zr}{}^{b} \end{array}\right]=\left[\begin{array}{ccc} \cos(A_{r}) & 0 & -\sin(A_{r})\\ 0 & 1 & 0\\ \sin(A_{r}) & 0 & -\cos(A_{r}) \end{array}\right]\left[\begin{array}{c} 0\\ 0\\ T_{r} \end{array}\right]\\ \left[\begin{array}{c} M_{xr}{}^{b}\\ M_{yr}{}^{b}\\ M_{zr}{}^{b} \end{array}\right]=\left[\begin{array}{ccc} 0 & z_{r} & y_{r}\\ -z_{r} & 0 & -x_{r}\\ -y_{r} & x_{r} & 0 \end{array}\right]\left[\begin{array}{c} T_{xr}{}^{b}\\ T_{yr}{}^{b}\\ T_{zr}{}^{b} \end{array}\right] \end{array}\right.$$

Similarly, the equations for the force and moment of the left rotor and the tail rotor can be expressed as follows:(5)$$\left\{ \begin{array}{l} \left[\begin{array}{c} T_{xl}{}^{b}\\ T_{yl}{}^{b}\\ T_{zl}{}^{b} \end{array}\right]=\left[\begin{array}{ccc} \cos(A_{l}) & 0 & -\sin(A_{l})\\ 0 & 1 & 0\\ \sin(A_{l}) & 0 & -\cos(A_{l}) \end{array}\right]\left[\begin{array}{c} 0\\ 0\\ T_{l} \end{array}\right]\\ \left[\begin{array}{c} M_{xl}{}^{b}\\ M_{yl}{}^{b}\\ M_{zl}{}^{b} \end{array}\right]=\left[\begin{array}{ccc} 0 & z_{l} & -y_{l}\\ -z_{l} & 0 & -x_{l}\\ y_{l} & x_{l} & 0 \end{array}\right]\left[\begin{array}{c} T_{xl}{}^{b}\\ T_{yl}{}^{b}\\ T_{zl}{}^{b} \end{array}\right] \end{array}\right.$$
(6)$$\left\{ \begin{array}{l} \left[\begin{array}{c} T_{xb}{}^{b}\\ T_{yb}{}^{b}\\ T_{zb}{}^{b} \end{array}\right]=\left[\begin{array}{ccc} 1 & 0 & 0\\ 0 & \cos(A_{b}) & -\sin(A_{b})\\ 0 & \sin(A_{b}) & \cos(A_{b}) \end{array}\right]\left[\begin{array}{c} 0\\ 0\\ T_{b} \end{array}\right]\\ \left[\begin{array}{c} M_{xb}{}^{b}\\ M_{yb}{}^{b}\\ M_{zb}{}^{b} \end{array}\right]=\left[\begin{array}{ccc} 0 & z_{b} & 0\\ -z_{b} & 0 & x_{b}\\ 0 & -x_{b} & 0 \end{array}\right]\left[\begin{array}{c} T_{xb}{}^{b}\\ T_{yb}{}^{b}\\ T_{zb}{}^{b} \end{array}\right] \end{array}\right.$$
where $T_{r}{}^{b}={\left[\begin{array}{ccc} T_{xr}{}^{b} & T_{yr}{}^{b} & T_{zr}{}^{b} \end{array}\right]}^{T}$, $T_{l}{}^{b}={\left[\begin{array}{ccc} T_{xl}{}^{b} & T_{yl}{}^{b} & T_{zl}{}^{b} \end{array}\right]}^{T}$ and $T_{b}{}^{b}={\left[\begin{array}{ccc} T_{xb}{}^{b} & T_{yb}{}^{b} & T_{zb}{}^{b} \end{array}\right]}^{T}$ are the thrust vectors expressed in the BBF, and $A_{r}$, $A_{l}$ and $A_{b}$ are the tilt angles of three rotors.

#### 2.3.2. Aerodynamic Model

According to the definition of coordinate systems and the description of the model, the aerodynamic forces and moments acting on the UAV are generally presented in the wind axis coordinate frame, as follows:(7)$$\begin{array}{l} C_{w}^{b}=\left[\begin{array}{ccc} \cos\alpha_{w} & 0 & -\sin\alpha_{w}\\ 0 & 1 & 0\\ \sin\alpha_{w} & 0 & \cos\alpha_{w} \end{array}\right]\left[\begin{array}{ccc} \cos\beta_{w} & -\sin\beta_{w} & 0\\ \sin\beta_{w} & \cos\beta_{w} & 0\\ 0 & 0 & 1 \end{array}\right]\\ P_{w}^{b}=\left[\begin{array}{ccc} 0 & -z_{w} & y_{w}\\ z_{w} & 0 & -x_{w}\\ -y_{w} & x_{w} & 0 \end{array}\right]\\ {\vec{F}}_{w}^{b}=\left[\begin{array}{c} F_{xw}{}^{b}\\ F_{xw}{}^{b}\\ F_{xw}{}^{b} \end{array}\right]=C_{w}^{b}\cdot \left[\begin{array}{c} -D_{w}\\ C_{w}\\ -L_{w} \end{array}\right]\\ {\vec{M}}_{w}^{b}=\left[\begin{array}{c} M_{xw}{}^{b}\\ M_{yw}{}^{b}\\ M_{zw}{}^{b} \end{array}\right]=C_{w}^{b}\cdot \left[\begin{array}{c} l_{w}\\ m_{w}\\ n_{w} \end{array}\right]+P_{w}^{b}\cdot \left[\begin{array}{c} F_{zw}{}^{b}\\ F_{zw}{}^{b}\\ F_{zw}{}^{b} \end{array}\right] \end{array}$$
where $C_{w}^{b}$ is the WAF-BFF transformation matrix, $\alpha_{w}$ is the angle of attack, $\beta_{w}$ is the angle of sideslip, ${\left(x_{w},y_{w},z_{w}\right)}^{T}$ is the coordinate of the pneumatic center of pressure relative to the center of gravity, ${\left(D_{w},C_{w},L_{w}\right)}^{T}$ are the aerodynamic forces, and ${\left(l_{w},m_{w},n_{w}\right)}^{T}$ are the aerodynamic moments.

#### 2.3.3. Gravity Model

In addition to the above external forces, the gravity based on BFF of the UAV can be expressed as:(8)$$\left[\begin{array}{c} F_{xg}\\ F_{yg}\\ F_{zg} \end{array}\right]=C_{n}^{b}\left[\begin{array}{c} 0\\ 0\\ mg \end{array}\right]$$

According to Equations (4)–(8), the combined force and moment of the UAV at the center of gravity can be presented as follows:(9)$$\left[\begin{array}{l} F_{x}{}^{b}\\ F_{y}{}^{b}\\ F_{z}{}^{b}\\ M_{x}{}^{b}\\ M_{y}{}^{b}\\ M_{z}{}^{b} \end{array}\right]=\left[\begin{array}{l} T_{xr}{}^{b}+T_{xl}{}^{b}+T_{xb}{}^{b}+F_{xw}{}^{b}+F_{xg}{}^{b}\\ T_{yr}{}^{b}+T_{yl}{}^{b}+T_{yb}{}^{b}+F_{yw}{}^{b}+F_{yg}{}^{b}\\ T_{zr}{}^{b}+T_{zl}{}^{b}+T_{zb}{}^{b}+F_{zw}{}^{b}+F_{zg}{}^{b}\\ M_{xr}{}^{b}+M_{xl}{}^{b}+M_{xb}{}^{b}+M_{xw}{}^{b}\\ M_{yr}{}^{b}+M_{yl}{}^{b}+M_{yb}{}^{b}+M_{yw}{}^{b}\\ M_{zr}{}^{b}+M_{zl}{}^{b}+M_{zb}{}^{b}+M_{zw}{}^{b} \end{array}\right]$$

### 2.4. Tilt Angle-Airspeed Envelope Model

In this section, a tilt angle-airspeed envelope model is established to provide flight path constraints for optimal trajectory planning. The modeling assumes that the acceleration of the BWTR is approximately zero in the takeoff–tilting transition process, and the UAV is in a steady-state flight state. In the takeoff–tilting transition process of the tilt-rotor UAV, if the airspeed of the UAV is too low, the wings will stall, while the maximum airspeed is limited by the effective power of the rotor. Therefore, the takeoff–tilting transition process of the UAV can only be in a certain range of airspeed.

Against the above, an analytical method for determining the tilt angle-airspeed envelope of the tilt-rotor UAV is presented in this paper. The method is based on the reasonable matching of rotor and wing aerodynamic force to establish the tilt angle-airspeed envelope model with wing stall and rotor available power as constraints to obtain the boundary of the tilting corridor of the tilt-rotor UAV, which are the flight path constraints of optimal trajectory planning.

#### 2.4.1. Trim Analysis

The trim analysis is based on the balance of the combined force and moment of the UAV, and appropriate mathematical methods are adopted to determine the control input variables and flight state variables required for the stable flight of the tilt-rotor UAV. The control input variables include the rotor thrust, tilt angle, deflection angle of the elevon, and the state variables include attitude angle and airspeed. The trim analysis of the UAV is very important for determining the boundary of the tilting corridor and optimal trajectory planning.

The trim analysis assumes that the tilt-rotor UAV flies steadily in a straight line and its angular velocity, angular acceleration and linear acceleration are zero. The constraints are expressed as follows:(10)$$\left\{ \begin{array}{l} \phi =\theta =\psi =0\\ p=q=r=0\\ \dot{u}=\dot{v}=\dot{w}=0 \end{array}\right.$$

According to Equations (9)–(10), the body motion equation of the tilt-rotor UAV can be given as:(11)$$\{\begin{array}{l} F_{x}{}^{b}=T_{xr}{}^{b}+T_{xl}{}^{b}+T_{xb}{}^{b}+F_{xw}{}^{b}+F_{xg}{}^{b}=0\\ F_{y}{}^{b}=T_{yr}{}^{b}+T_{yl}{}^{b}+T_{yb}{}^{b}+F_{yw}{}^{b}+F_{yg}{}^{b}=0\\ F_{z}{}^{b}=T_{zr}{}^{b}+T_{zl}{}^{b}+T_{zb}{}^{b}+F_{zw}{}^{b}+F_{zg}{}^{b}=0\\ M_{x}{}^{b}=M_{xr}{}^{b}+M_{xl}{}^{b}+M_{xb}{}^{b}+M_{xw}{}^{b}=0\\ M_{y}{}^{b}=M_{yr}{}^{b}+M_{yl}{}^{b}+M_{yb}{}^{b}+M_{yw}{}^{b}=0\\ M_{z}{}^{b}=M_{zr}{}^{b}+M_{zl}{}^{b}+M_{zb}{}^{b}+M_{zw}{}^{b}=0 \end{array}$$

The specific calculation method for the force and moment in Equation (11) is described in Section 2.3. Equation (11) is a nonlinear system of equations; its solution needs a numerical calculation method.

At present, the main methods for solving nonlinear equations include the Simplex method, Newton–Raphson method, gradient descent method [26], etc. The Newton–Raphson method is used to solve the body motion equation for the tilt-rotor UAV, which has the characteristics of fast convergence speed and low requirement on initial value.

#### 2.4.2. Tilting Corridor

The lower boundary of the tilting corridor is the minimum airspeed of the UAV without the wing stall at different tilt angles while satisfying the balance of lift force, drag force, thrust, and gravity in the takeoff–tilting transition process. Besides, the gravity of the UAV is mainly balanced by the rotor thrust force to wing lift force in the takeoff–tilting transition process. According to Equation (7), the lift force provided by the wing is limited by the critical stall angle of attack of the wing. Therefore, the critical stall angle of attack can be used to calculate the lower boundary of the tilting corridor.

The upper boundary of the tilting corridor is the maximum airspeed of the UAV limited by the effective power of the rotor at different tilt angles while satisfying the balance of force acting on the UAV in the takeoff–tilting transition process. Therefore, the effective power of the rotor can be used to determine the upper boundary of the tilting corridor, which can be obtained according to Equation (3).

Figure 5 shows the tilt-rotor corridor boundary of the BWTR under different combinations of tilt angles and speeds, which is obtained through the tilt angle-airspeed envelope model. Consequently, the conclusions are as follows:The takeoff–tilting transition process of the UAV can be completed by different combinations of the angle of attack of the fuselage, the tilt angle of the rotor and airspeed;As the UAV transitions from the lower boundary of the tilting corridor to the upper boundary, the fuselage angle of orientation also changes from the wing stall angle of orientation.In order to ensure the safety and efficiency of the UAV, the optimal trajectory planning should treat the tilting corridor as a constraint condition.

## 3. Optimal Take-Off Trajectory Planning

The optimal trajectory planning for takeoff of the tilt-rotor UAV can be described as: finding an optimal trajectory from a cluster of takeoff trajectories, so that a certain performance evaluation index is optimal when the UAV transitions from the initial state to the target state. Different from the general optimization problem, both flight state and performance indexes are functions of time and space in the takeoff–tilting transition process. In consequence, the optimal trajectory planning for takeoff of the tilt-rotor UAV can be summarized as a nonlinear dynamic optimal control problem with flight state constraints and control constraints.

As a branch of modern control theory, optimal control can effectively solve the control optimization problems of high-order, multi-input, and multiple-output systems. Mathematically, the optimal control problem can be described as: To determine the extreme value of the performance index with state vector and control vector as variables, according to the motion equation and constraint conditions.

Formulations of optimal control problems refer to the equation of state, the constraints of state variables and control variables, and performance index functions.

### 3.1. Optimal Control Problem

#### 3.1.1. Equation of State

The equation of state of the controlled system is supposed as follows:(12)$$\begin{array}{l} \dot{X}\left(t\right)=f\left[X\left(t\right),U\left(t\right)\right]\\ X\left(t\right)=\left[V_{z}{}^{n}\left(t\right),V_{x}{}^{n}\left(t\right),Z_{n}\left(t\right),A_{p}\left(t\right)\right]\\ U\left(t\right)=\left[T_{rl}\left(t\right),A_{rl}\left(t\right),\alpha_{w}\left(t\right)\right] \end{array}$$
where X(t) is the state variable and U(t) is the control variable. The initial and terminal constraints of the state variable can be expressed as: $X\left(t_{0}\right)=X_{0}$ and $X\left(t_{f}\right)\in S$, where S is a set of target states. $V_{z}{}^{n}$ is the NED-based vertical velocity, $V_{x}{}^{n}$ is the NED-based forward velocity, $Z_{n}$ is the NED-based vertical position, $A_{p}$ is the track angle, $T_{rl}$ is the resultant force of left and right rotors and $A_{rl}$ is the tilt angle of the rotors.

The equation of state of optimal control can be expressed by the kinematic model established in Section 2.3. Besides, it is also based on the model assumptions satisfying the constraints. The model assumptions are given as follows:The optimal trajectory planning is used to determine the optimal takeoff scheme for the UAV. The interference and parameter uncertainty factors do not affect the comparison results for optimal trajectory planning. The acceleration is mainly provided by the thrust of the rotor and the aerodynamic forces of the body during the takeoff–tilting transition process. Therefore, the UAV is regarded as an idealized model.During the tilt transition phase, the longitudinal trajectory of the UAV is an essential parameter that affects its safety and stability, and the UAV is considered as an ideal model. Therefore, this manuscript focuses on optimal planning of longitudinal trajectories and studies the longitudinal equations of motion of the UAV. In the takeoff–tilting process, the equation of state of the UAV, which does not contain transverse state variables and satisfies the horizontal no-slip flight condition, can be expressed as follows:
(13)β=ϕ=ψ=p=r=0

3.According to Section 2.4, when the UAV is in the tilting corridor, it can maintain the longitudinal pitching moment balance through the control moment provided by tail rotor tension and elevon deflection, and the trajectory motion and attitude motion of the UAV are decoupled. Therefore, the equation of state of the UAV can only consider the trajectory motion state variables.

In conclusion, Equation (10) can be expressed as the equation of state of the UAV as follows:(14)$$\left\{ \begin{array}{l} {\dot{V}}_{x}{}^{n}=\frac{1}{m}\left\{T_{rl}\sin(\frac{\pi }{2}-A_{rl}+\alpha_{w}+A_{p})+L_{w}\cos(A_{p})-D_{w}sin(A_{p})\right\}-g\\ {\dot{V}}_{z}{}^{n}=\frac{1}{m}\left\{T_{rl}\cos(\frac{\pi }{2}-A_{rl}+\alpha_{w}+A_{p})-L_{w}sin(A_{p})-D_{w}cos(A_{p})\right\}\\ A_{p}=\arctan\left(\frac{V_{z}{}^{n}}{V_{x}{}^{n}}\right)\\ {\dot{Z}}^{n}=V_{z}{}^{n} \end{array}\right.$$

In addition, the equation of state is also different in various takeoff–tilting schemes.

#### 3.1.2. Constraints

In the takeoff–tilting process of the UAV, the initial and terminal constraints of optimal trajectory planning should be determined first. Secondly, the path constraints of the UAV are determined, including control variable constraints and state variable constraints. Finally, it is necessary to comprehensively consider the control margin of the servo actuator of the UAV and formulate reasonable constraint conditions.

The longitudinal trajectory of UAVs mainly includes forward and vertical positions and track angle., and the angle of attack needs to meet the control requirements of the UAV pitch channel under this constraint. Therefore, the NED-based initial and terminal constraints of the takeoff–tilting process can be expressed as follows:(15)$$\left\{ \begin{array}{l} V_{x}{}^{n}\left(t_{0}\right)=V_{h0},V_{x}{}^{n}\left(t_{f}\right)=V_{hf}\\ V_{z}{}^{n}\left(t_{0}\right)=V_{v0},V_{z}{}^{n}\left(t_{f}\right)=V_{vf}\\ Z^{n}\left(t_{0}\right)=Z_{0},Z_{n}\left(t_{f}\right)=Z_{f}\\ \alpha_{w}\left(t_{0}\right)=\alpha_{0},\alpha_{w}\left(t_{f}\right)=\alpha_{f}\\ A_{p}\left(t_{0}\right)=A_{0},A_{p}\left(t_{f}\right)=A_{f} \end{array}\right.$$
where $t_{0}$ and $t_{f}$ are the initial and terminal time, $V_{v0}$ and $V_{vf}$ are the initial and terminal vertical velocity, $V_{h0}$ and $V_{hf}$ are the initial and terminal horizontal velocity, $Z_{0}$ and $Z_{f}$ are the initial and terminal height, $\alpha_{0}$ and $\alpha_{f}$ are the initial and terminal angle of attack, $A_{0}$ and $A_{f}$ are the initial and terminal angle of track.

The thrust of the left and right rotors and tilt angle are actively controlled in the whole trajectory of the UAV. Therefore, it is necessary to determine the constraint conditions of the thrust of rotor and tilt angle. Besides, according to Section 2.4, the angle of attack constraint of the UAV is determined according to the upper and lower boundaries of the tilting corridor. In addition, the state variables of the UAV, including the forward and vertical velocity, altitude angles and flight path angle based on NED, should also meet the constraints of the tilting corridor.

To sum up, in order to make the flight path of the UAV smooth in the takeoff–tilting process and control the motion attitude within the controllable range of each actuator, the path constraints of the UAV can be expressed as follows:

Path constraints on state variables include,
(16)$$\left\{ \begin{array}{l} V_{v\min}\leq V_{z}{}^{n}\left(t\right)\leq V_{v\max}\\ V_{h\min}\leq V_{x}{}^{n}\left(t\right)\leq V_{h\max}\\ Z_{\min}\leq Z_{n}\left(t\right)\leq Z_{\max}\\ A_{p\min}\leq A_{p}\left(t\right)\leq A_{p\max} \end{array}\right.$$

Path constraints on control variables include,
(17)$$\left\{ \begin{array}{l} T_{\min}\leq T_{rl}\left(t\right)\leq T_{\max}\\ A_{\min}\leq A_{rl}\left(t\right)\leq A_{\max}\\ \alpha_{w\min}\leq \alpha_{w}\left(t\right)\leq \alpha_{w\max} \end{array}\right.$$

In the takeoff–tilting process, the servo actuators of the UAV participate in attitude control. For example, the longitudinal unbalanced torque of the UAV caused by the main rotor tilting needs to be balanced by the tail rotor and elevons. Therefore, constraining the servo actuators according to the control margin makes the optimal trajectory planning physically meaningful. Constraints of servo actuators are given as follows:(18)$$\left\{ \begin{array}{l} 0\leq T_{b}\left(t\right)\leq T_{b\max}\\ A_{b\min}\leq A_{b}\left(t\right)\leq A_{b\max}\\ A_{ar\min}\leq A_{ar}\left(t\right)\leq A_{ar\max}\\ A_{ar\min}\leq A_{al}\left(t\right)\leq A_{ar\max} \end{array}\right.$$
where $T_{b}$ and $A_{b}$ are the thrust and tilt angle of the tail rotor, respectively, $A_{al}$ and $A_{ar}$ are the deflection angle of the left and right elevons, respectively.

#### 3.1.3. Performance Index

The performance index of the optimal control problem generally has the form as follows:(19)$$J=\phi \left[X\left(t_{f}\right),t_{f}\right]+\int_{t_{0}}^{t_{f}}L\left[X\left(t\right),U\left(t\right),t\right]dt$$

According to Equation (19), the performance index consists of an integral index and terminal index. The form of the performance index needs to be determined according to the actual problem. The Mayer problem can be represented by the terminal index, while the Lagrange problem can be represented by the integral index.

The structure and flight mode of the tilt-rotor UAV change during the takeoff–tilting process. On the one hand, the structure and aerodynamic parameters of the UAV are changed due to the tilting of the rotors. On the other hand, the switching of control strategy changes the control effect of the actuator. Therefore, for the sake of safety, the time of the whole takeoff–tilting process should be optimized to the minimum that the control ability allows. In addition, the takeoff–tilting process of the tilt-rotor UAV is a process in which kinetic energy and potential energy increase simultaneously, and the main energy source of the UAV is the main rotor. Therefore, another purpose of optimal trajectory planning is to minimize the energy consumption.

In conclusion, the optimal control problem is formulated to minimize the total time and energy consumption of the takeoff–tilting process of the UAV.

Mathematically, the performance indexes are defined as follows:(20)$$J=K_{t}t_{f}+\frac{1}{t_{f}-t_{0}}\int_{t_{0}}^{t_{f}}P\left(t\right)dt$$
where, P is the power of the main rotors, which can be obtained according to Section 2.3., and $K_{t}$ is the time factor scaling coefficient.

In this paper, the main aim is to find an optimal take-off scheme suitable for the novel tilt-rotor UAV. Therefore, we investigate the trajectories and performance parameters of the three take-off schemes, where the energy consumption of the whole process is one of the performance parameters. According to the rotor model in Section 2.3.1, we have obtained the required power of the rotors under different thrusts, as shown in Equation (3). Besides, the primary energy source of the UAV is the work done by the rotor, so the energy consumption is approximately equal to the superposition of the power required by the rotor during the entire takeoff process.

#### 3.1.4. Optimization Methods

The takeoff–tilting process of the BWTR is complicated, with many control variables and coupling of state variables. Furthermore, the process has strict constraints of starting terminal, flight path and actuator. Its performance index is also strongly nonlinear. Therefore, the optimal control problem may defy analytical solutions. In this paper, direct collocation nonlinear programming (DCNLP) is a highly effective numerical solution approach to convert the dynamic optimal control problem into a static parameter optimization problem. In particular, the direct collocation method converts both the state variables and control variables into parameters as optimization objects.

The direct collocation method begins with the definition of a series of n time points within the solution time interval $\left[t_{0}:t_{f}\right]$ of the optimal control problem. For i=1,2,…,n, define
(21)$$\overset{n}{\overbrace{t_{0},\ldots ,t_{i},\ldots ,t_{n-1},t_{f}}}$$

As shown in Figure 6, the state variables and control variables at each time point can be used as a group of corresponding discretized solution parameters that can be represented as follows:(22)$$\begin{array}{l} X_{0},\ldots ,X_{i},\ldots ,X_{n-1},X_{f}\\ U_{0},\ldots ,U_{i},\ldots ,U_{n-1},U_{f} \end{array}$$
where $X_{i}$ are the discretized parameters of the state vector $X=[V_{x}{}^{n},\;V_{z}{}^{n},\;Z^{n},\;A_{p}]$ at the time point of $t_{i}$, $U_{i}$ are the discretized parameters of the control vector $U=[T_{rl},\;A_{rl},\;\alpha_{w}]$ at the time point of $t_{i}$. Specifically, polynomials can be used between time points to fit the variation rule of state variables with time. Additionally, constraints of state variables and control variables can be imposed on each time point.

Putting all of the discretized variables together and including the open final time:(23)$$C=[{\left(X,U,U_{m}\right)}_{1},{\left(X,U,U_{m}\right)}_{2},\cdots {\left(X,U,U_{m}\right)}_{i}\cdots {\left(X,U,U_{m}\right)}_{n-1},{\left(X,U\right)}_{t_{f}},t_{0},t_{f}]$$
where $U_{mi}$ is the control variable of the middle point of two discrete time points.

Using Simpson’s one-third rule to integrate the equation of state between two discrete time points, the resulting equality constraints can be expressed as:(24)$$\begin{array}{l} X_{i+1}=X_{i}+\frac{\Delta t}{6}\left[f\left(X_{i},U_{i},t_{i}\right)+4f\left(X_{mi},U_{mi},t_{mi}\right)+f\left(X_{i+1},U_{i+1},t_{i+1}\right)\right]\\ \Delta t=(t_{f}-t_{0})/(n-1) \end{array}$$

According to Equation (24), the discrete performance index function can be presented as follows:(25)$$J=t_{n}+\frac{1}{6}\sum_{i=1}^{n-1}\left[P\left(t_{i}\right)+4P\left(t_{mi}\right)+P\left(t_{i+1}\right)\right]$$

The path constraints in Equations (16)–(18) are enforced at the series of discrete time points as bounds on the solution parameters. The fixed initial and terminal conditions in Equation (15) are enforced by equating upper and lower bounds on the corresponding variables. Then, the optimal control problem can be solved efficiently using the direct collocation method.

### 3.2. Trajectory Planning Schemes

#### 3.2.1. Optimal Planning in VTO

The tilt-rotor UAV has the advantage of realizing both VTOL and cruise flight modes. Therefore, in the optimal trajectory planning for the takeoff–tilting process, the VTO scheme is considered first. Figure 7 shows the flight path of the VTO scheme, whose trajectory can be described as follows: First, the UAV takes off vertically from the initial height to the terminal height. Second, the UAV maintains the flight altitude, while the tilt angle changes from 0 to 90 deg. Meanwhile, the airspeed of the UAV gradually increases to balance gravity, thrust vectoring, lift, and drag. Finally, the UAV flies at a constant altitude and heading with the identical airspeed and tilt angle.

The flight path of the VTO scheme is divided into two stages: vertical and tilting. Figure 8 shows the force diagram of the UAV in the vertical and tilting stages of the VTO. According to Figure 8, the state equation of the UAV in the VTO can be established. In particular, the equation of state as (14) of the vertical stage can be further expressed as:(26)$$\left\{ \begin{array}{l} {\dot{V}}_{x}{}^{n}=0\\ {\dot{V}}_{z}{}^{n}=\left(T_{rl}-D_{w}-mg\right)/m\\ A_{p}=90\\ {\dot{Z}}^{n}=V_{z}{}^{n} \end{array}\right.$$

The state equation for the tilting stage can be given as:(27)$$\left\{ \begin{array}{l} {\dot{V}}_{x}{}^{n}=\left(T_{rl}\cos(\pi /2-A_{rl}+\alpha_{w})-D_{w}\right)/m\\ {\dot{V}}_{z}{}^{n}=0\\ A_{p}=0\\ {\dot{Z}}^{n}=V_{z}{}^{n} \end{array}\right.$$

The initial, terminal, and path constraints for the vertical stage are shown as:(28)$$\begin{array}{l} \left\{ \begin{array}{l} V_{h0}=V_{hf}=0\;m/s\;V_{v0}=V_{vf}=0\;m/s\;Z_{0}=0\;m,Z_{f}=40\;m\;\\ \alpha_{0}=\alpha_{f}=0\;\deg\;A_{0}=A_{f}=0\;\deg \end{array}\right.\\ \left\{ \begin{array}{l} V_{v\min}=0\;m/s,V_{v\max}=5\;m/s\;V_{h\min}=V_{h\max}=0\;m/s\\ A_{p\min}=0\;\deg,A_{p\max}=90\;\deg\;Z_{\min}=0\;m,Z_{\max}=40\;m \end{array}\right.\\ \{T_{\min}=0\;\mathrm{kg},T_{\max}=80\;\mathrm{kg}\;A_{\min}=A_{\max}=0\;\deg\;\alpha_{w\min}=\alpha_{w\max}=0\;\deg \end{array}$$

The initial, terminal, and path constraints for the tilting stage are shown as:(29)$$\begin{array}{l} \left\{ \begin{array}{l} V_{h0}=0\;m/s,V_{hf}=33\;m/s\;V_{v0}=V_{vf}=0\;m/s\;Z_{0}=Z_{f}=40\;m\;\\ \alpha_{0}=0\;\deg\;\alpha_{f}=2\;\deg\;A_{0}=A_{f}=0\;\deg \end{array}\right.\\ \left\{ \begin{array}{l} V_{v\min}=V_{v\max}=0\;m/s\;V_{h\min}=0\;m/s,V_{h\max}=33\;m/s\\ A_{p\min}=A_{p\max}=0\;\deg\;Z_{\min}=Z_{\max}=40\;m \end{array}\right.\\ \left\{ \begin{array}{l} T_{\min}=0\;\mathrm{kg},T_{\max}=80\;\mathrm{kg}\;A_{\min}=0\;\deg,A_{\max}=90\;\deg\\ \alpha_{w\min}=0\;\deg,\alpha_{w\max}=2\;\deg \end{array}\right. \end{array}$$

#### 3.2.2. Optimal Planning in STO

In order to save fuel and increase the range and maximum takeoff weight, the tilt-rotor UAV can also take off at a short distance if the site conditions permit. Compared with the vertical takeoff and landing scheme, the short take-off scheme brings better stability and wind resistance for the UAV and has the advantages of avoiding adverse ground effects, which can effectively improve the payload and safety of the UAV. Therefore, it is of great significance.

The tilt-rotor UAV achieves short take-off by tilting the rotors to provide forward thrust vectoring. In this paper, the optimal trajectory planning for the UAV in the VTO scheme is carried out for taxiing on a limited field and fast and stable climbing after take-off.

Figure 9 shows the flight path of the STO scheme, whose trajectory can be described as follows: First, the rotors of the UAV tilt an initial constraint angle. Then, the thrust vectoring of the rotors provides the UAV with a forward component that acts in the same way as the vector thrust of conventional aircraft. The UAV accelerates to the takeoff speed that generates enough aerodynamic lift. The lift and the vertical components of thrust work together to lift the UAV off the ground. Finally, after the UAV takes off, the rotors continue to tilt to the horizontal position, while the vertical and forward velocities continue to accumulate, and finally meet the terminal constraints.

The flight path of the STO scheme is divided into two stages: taxiing and takeoff. Figure 10 shows the force diagram of the UAV in taxiing and takeoff stages of the STO.

In the taxiing stage, the forward acceleration of the UAV is provided by the forward component of the rotor thrust to overcome the resistance and friction. The lift force, the vertical component of the thrust and the support force of the UAV balance the gravity together. When the support force of the UAV gradually decreases to zero, the UAV enters the take-off stage. In the take-off stage, the UAV has a large initial forward velocity. In particular, the forward acceleration is provided by the forward component of the rotor thrust vectoring, while the vertical acceleration is provided by the increasing vertical component of the thrust and the lift force of the UAV.

According to Figure 10, the state equation of the UAV in the STO can be established. In particular, the equation of state as (14) of the taxiing stage can be further expressed as:(30)$$\left\{ \begin{array}{l} {\dot{V}}_{x}{}^{n}=\left(T_{rl}\cos(\pi /2+\alpha_{w}-A_{rl})+N\cdot \mu -D_{w}\right)/m\\ {\dot{V}}_{z}{}^{n}=\left(T_{rl}\sin(\pi /2+\alpha_{w}-A_{rl})+N+L_{w}-mg\right)/m\\ A_{p}=0\\ {\dot{Z}}^{n}=V_{z}{}^{n} \end{array}\right.$$

The state equation for the takeoff stage can be given as:(31)$$\left\{ \begin{array}{l} {\dot{V}}_{x}{}^{n}=\left(T_{rl}\cos(\pi /2-A_{rl}+\alpha_{w}+A_{p})-L_{w}sin(A_{p})-D_{w}cos(A_{p})\right)/m\\ {\dot{V}}_{z}{}^{n}=\left(T_{rl}\sin(\pi /2-A_{rl}+\alpha_{w}+A_{p})+L_{w}\cos(A_{p})-D_{w}sin(A_{p})-mg\right)/m\\ A_{p}=\arctan\left(\frac{V_{z}{}^{n}}{V_{x}{}^{n}}\right)\\ {\dot{Z}}^{n}=V_{z}{}^{n} \end{array}\right.$$

The supporting force is included in the state equation of the UAV in the taxiing stage, which is taken as a control variable in the optimal trajectory planning. Therefore, the initial, terminal, and path constraints for the taxiing stage are shown as:(32)$$\begin{array}{l} \left\{ \begin{array}{l} V_{h0}=0\;m/s\;,V_{hf}=10\;m/s\;V_{v0}=V_{vf}=0\;m/s\;Z_{0}=Z_{f}=0\;m\;\\ \alpha_{0}=\alpha_{f}=2\;\deg\;\;\;A_{0}=A_{f}=0\;\deg \end{array}\right.\\ \left\{ \begin{array}{l} V_{v\min}=V_{v\max}=0\;m/s\;V_{h\min}=0\;m/s,V_{h\max}=10\;m/s\\ A_{p\min}=A_{p\max}=0\;\deg\;Z_{\min}=Z_{\max}=0\;m \end{array}\right.\\ \{T_{\min}=0\;\mathrm{kg},T_{\max}=80\;\mathrm{kg}\;A_{\min}=A_{\max}=15\;\deg\;\alpha_{w\min}=\alpha_{w\max}=2\;\deg \end{array}$$

The initial, terminal, and path constraints for the takeoff stage are shown as:(33)$$\begin{array}{l} \left\{ \begin{array}{l} V_{h0}=10\;m/s,V_{hf}=33\;m/s\;V_{v0}=V_{vf}=0\;m/s\;Z_{0}=0\;m,Z_{f}=40\;m\;\\ \alpha_{0}=2\;\deg\;\alpha_{f}=0\;\deg\;\;A_{0}=A_{f}=0\;\deg \end{array}\right.\\ \left\{ \begin{array}{l} V_{v\min}=0\;m/s\;V_{v\max}=5\;m/s\;V_{h\min}=10\;m/s,V_{h\max}=33\;m/s\\ A_{p\min}=0\;\deg,A_{p\max}=15\;\deg\;\;Z_{\min}=0\;m,Z_{\max}=40\;m \end{array}\right.\\ \left\{ \begin{array}{l} T_{\min}=0\;\mathrm{kg},T_{\max}=80\;\mathrm{kg}\;\;\;\;A_{\min}=0\;\deg,A_{\max}=90\;\deg\\ \alpha_{w\min}=-5\;\deg,\alpha_{w\max}=7\;\deg \end{array}\right. \end{array}$$

When the terminal state constraints are met, different combinations of takeoff airspeed and tilt angle correspond to different taxiing distances, energy consumption and time, so the STO scheme that is most suitable for flight conditions can be selected according to comparative analysis.
(34)$$\left\{ \begin{array}{l} V_{at}=10\;m/s\\ A_{rlt}=75^{\circ } \end{array}\right.$$
where $V_{at}$ is the takeoff velocity and $A_{rlt}$ is the fixed tilt angle in taxiing stage.

#### 3.2.3. Optimal Planning in TTO

The simulation results for the STO scheme show that the energy consumption and time of the takeoff–tilting process decrease with the increase in takeoff velocity. In addition, the maximum thrust of the tilt-rotor UAV is larger than the force of gravity, so the TTO scheme is proposed, that is, when the UAV accumulates airspeed in the tilting process, it also climbs higher, making full use of aerodynamic lift to save energy. Furthermore, in order to optimize the time and energy consumption of the takeoff–tilting process, the optimal trajectory planning is also carried out.

Due to the characteristics of the motor, the UAV cannot directly balance the state of gravity and lift, so the TTO scheme starts from hovering on the ground. Figure 11 shows the flight path of the TTO scheme, whose trajectory can be described as follows: First, the thrust of the rotors increases until it balances the gravity of the UAV while the UAV hovers over the ground. Then, the rotor tilts and the UAV generates a forward velocity vector. At this point, the forward component of rotor thrust vectoring provides forward acceleration, while the vertical component of rotor thrust vectoring and the lift force provides vertical acceleration for the UAV. Finally, the state variables of the UAV satisfy the terminal constraint condition.

Figure 12 shows the force diagram of the UAV in the VTO scheme. According to Figure 12, the equation of state as (14) of the TTO scheme can be further expressed as:(35)$$\left\{ \begin{array}{l} {\dot{V}}_{x}{}^{n}=\left(T_{rl}\cos(\pi /2-A_{rl}+\alpha_{w}+A_{p})-L_{w}sin(A_{p})-D_{w}cos(A_{p})\right)/m\\ {\dot{V}}_{z}{}^{n}=\left(T_{rl}\sin(\pi /2-A_{rl}+\alpha_{w}+A_{p})+L_{w}\cos(A_{p})-D_{w}sin(A_{p})-mg\right)/m\\ A_{p}=\arctan\left(\frac{V_{z}{}^{n}}{V_{x}{}^{n}}\right)\\ {\dot{Z}}^{n}=V_{z}{}^{n} \end{array}\right.$$

The initial, terminal, and path constraints for the TTO scheme are shown as:(36)$$\begin{array}{l} \left\{ \begin{array}{l} V_{h0}=V_{hf}=0\;m/s\;\;\;V_{v0}=0\;m/s,V_{vf}=33\;m/s\;\;\;Z_{0}=0\;m,Z_{f}=40\;m\;\\ \alpha_{0}=0\;\deg,\alpha_{f}=2\;\deg\;\;\;\;A_{0}=A_{f}=0\;\deg \end{array}\right.\\ \left\{ \begin{array}{l} V_{v\min}=0\;m/s,V_{v\max}=5\;m/s\;\;\;V_{h\min}=0\;m/s,V_{h\max}=33\;m/s\\ A_{p\min}=0\;\deg,A_{p\max}=15\;\deg\;\;Z_{\min}=0\;m,Z_{\max}=40\;m \end{array}\right.\\ \left\{ \begin{array}{l} T_{\min}=0\;\mathrm{kg},T_{\max}=80\;\mathrm{kg}\;\;A_{\min}=0\;\deg,A_{\max}=90\;\deg\\ \alpha_{w\min}=-5\;\deg,\alpha_{w\max}=5\;\deg \end{array}\right. \end{array}$$

## 4. Simulation Results

In this paper, gradients for both the objective functions and the nonlinear constraints are calculated numerically. Different takeoff–tilting schemes, which result in different equations of state and path constraints, are investigated. In obtaining numerical solutions, constant initial and terminal conditions for the state and control variables are used.

### 4.1. Simulation Results in VTO

As shown in Figure 13, Figure 14 and Figure 15, the simulation results of the optimal trajectory planning for the UAV in the takeoff–tilting process, which applies to the VTO scheme, are presented to demonstrate the availability of the direct collocation method. The total time of optimal trajectory planning in the VTO scheme is 23.45 s, including 15.15 s in the vertical stage and 8.3 s in the tilting stage.

Figure 13 shows the time-varying state variables of the UAV in the optimal trajectory planning, including the acceleration, velocity, and position in the forward and vertical direction based on the NED. At the end of the vertical stage, the UAV hovers at a fixed height, and at the end of the tilting stage, the UAV flies at a constant velocity, height and heading. In the terminal state of the UAV, the forward position is 154.3 m, the forward velocity is 33.19 m/s, the vertical height is 40 m, and the vertical velocity is 0 m/s.

Figure 14 shows the time-varying control variables of the UAV in the optimal trajectory planning, which include the total thrust of left and right rotors, tilt angles of left and right rotors, and angle of attack of the UAV. Among them, the tilt angles of the left and right rotors begin to change at the beginning of the tilting stage. Additionally, the thrust of the main rotor increases first and then decreases, which is due to the insufficient accumulation of the airspeed of the UAV in the early stage of tilting, and the gravity is mainly balanced by the thrust of the rotor. In the late stage of tilting, the lift force of the wing is enough to balance the gravity, and the thrust of the rotor only needs to overcome the resistance. In addition, the lift force of the wing increases gradually in the tilting stage, so the angle of attack is selected as the control variable to ensure that the wing has sufficient lift efficiency.

Figure 15 shows the time-varying power and energy consumption of the UAV in VTO. The total energy consumption of the UAV is 265.3 kJ.

### 4.2. Simulation Results in STO

Taking the conditions shown in Equation (39) as an example, the simulation results of the optimal trajectory planning for the UAV in the takeoff–tilting process, which applies to the STO scheme, are presented as shown in Figure 16, Figure 17 and Figure 18. The total time of optimal trajectory planning in the STO scheme is 19.6 s, including 4.6 s in the taxiing stage and 15 s in the takeoff stage.

Figure 16 shows the time-varying state variables of the UAV in the optimal trajectory planning. At the end of the taxiing stage, the UAV taxis at a fixed takeoff velocity and tilt angle, while at the end of the takeoff stage, the UAV flies at a constant velocity, height and heading. The terminal states of the UAV are the same as those of the VTO scheme, except that the forward position is 371.2 m.

Figure 17 shows the time-varying control variables of the UAV in the optimal trajectory planning in STO. The tilt angles of the left and right rotor begin to change at the beginning of the takeoff stage. Unlike the VTO scheme, the thrust of the main rotor gradually decreases, which is because, during the taxiing stage, the aerodynamic lift of the UAV increases with time, and gravity is already balanced by lift force and the thrust of the rotors in the takeoff stage. In addition, the support force of the UAV is always zero because of the sufficient thrust of the rotors.

Figure 18 shows the time-varying power and energy consumption of the UAV in STO. The total energy consumption of the UAV is 254.6 kJ.

The above simulations only simulate the STO scheme in one case. In fact, however, the initial rotor tilt angle $A_{rl}$ and airspeed at the beginning of the takeoff stage $V_{a}$ are two crucial factors that affect the trajectory of the STO scheme. In order to explore the influence of factors $A_{rl}$ and $V_{a}$ on the time consuming, energy consuming and state variables of the STO scheme and to find the trajectory with the optimal performance parameters, the optimal trajectory planning was performed for the short takeoff scheme with different combinations of factors $A_{rl}$ and $V_{a}$, and the simulation results were analyzed statistically. The results of optimal trajectory planning are shown in Table 3 for different combinations of tilt angle and takeoff velocity. The value ranges of tilt angle and takeoff velocity are presented as follows:(37)$$\begin{array}{l} 50^{\circ }\leq A_{rl}\leq 85^{\circ }\\ 5\;m/s\leq V_{a}\leq 25\;m/s \end{array}$$
where $L_{ti}$ is sliding distance, $t_{ti}$ and $Q_{ti}$ are time and energy consumption in taxiing stage, $t_{tf}$ and $Q_{tf}$ are time and energy consumption in takeoff stage, and $t_{com}$ and $Q_{com}$ are total time and energy consumption.

The data in Table 3 are plotted in the form of a histogram to analyze the changes in various variables in the STO scheme over time under the combination of different tilt angles and takeoff velocities.

As shown in Figure 19, the taxiing distance decreases with the increase in the tilt angle and increases with the increase in the fixed takeoff velocity. In addition, the energy consumption decreases with the increase in the tilt angle and increases with the increase in the fixed takeoff velocity. Further, the time in the taxiing stage decreases with the increase in the tilt angle and increases with the increase in the fixed takeoff velocity. On the one hand, there is little difference in the time for different combinations because it is mainly limited by the maximum vertical velocity. On the other hand, the energy consumption in the takeoff stage decreases with the increase in takeoff velocity, while the tilt angle has little influence on it, because the takeoff stage is a process of increasing kinetic energy and potential energy, and the potential energy is confirmed by terminal state constraints. Under the same initial kinetic energy, different tilt angles only affect the constraints of the actuator, which is within the tilt-rotor corridor of the UAV. Therefore, the energy consumption is not affected.

The total energy consumption and time of the STO scheme are shown in Figure 20. For the UAV, it can be concluded that: when the tilt angle is smaller or the takeoff velocity is smaller, the total energy consumption and time of the UAV are smaller, and the taxiing distance is shorter.

### 4.3. Simulation Results in TTO

As shown in Figure 21, the simulation results of the optimal trajectory planning for the UAV in the TTO scheme are presented to demonstrate the superiority of the scheme. The total time of optimal trajectory planning in the TTO scheme is 11 s.

Figure 21 shows the time-varying state variables of the UAV in the TTO scheme. The forward velocity and vertical velocity based on NED change smoothly with time. Additionally, all of the state variables satisfy the initial and terminal constraints. The forward position is 228 m.

Figure 22 shows the time-varying control variables of the UAV in the TTO scheme. The lift force of the wing increases gradually with the increase in airspeed, which causes the thrust of the UAV to decrease gradually.

Figure 23 shows the time-varying power and energy consumption of the UAV in TTO. The total energy consumption of the UAV is 199.5 kJ.

### 4.4. Comparison of Taking-Off Schemes

As shown in Figure 24, the optimal control problem is transformed into a nonlinear programming problem by using the direct collocation method, and the optimal trajectory in different schemes is obtained. Under the same constraints, the VTO scheme has the advantages of minimum space occupation and simple requirements during takeoff. However, at the same time, a smaller payload, high power demand, poor wind resistance, energy consumption and service time are the shortcomings of this scheme. Compared with the VTO scheme, on the one hand the STO scheme has the advantages of less energy consumption and time, better stability, and wind resistance, which can effectively improve the payload of the UAV, but on the other hand, it needs to occupy the runway for pre-takeoff taxiing. Among all schemes, the TTO scheme consumes the least energy and takes the least time. It not only has all of the advantages of the STO scheme, but also does not need to occupy the runway for taxiing.

## 5. Conclusions

In this paper, an innovative tilt-rotor UAV was presented, and its dynamic model was established considering the effect of axial air flow velocity. Additionally, an analytical method of determining the tilt angle-airspeed envelope of a tilt-rotor UAV was presented, so that the constraints of optimal trajectory planning could be obtained.

A novel takeoff–tilting TTO scheme was proposed and researched. The optimal trajectory planning results for three takeoff–tilting schemes were compared and analyzed, and the optimal trajectory, including state variables and control variables in the entire takeoff–tilting process, was obtained.

Simulation results demonstrate that the time consumption of the TTO scheme is 47% of that of the VTO scheme, and the energy consumption of the TTO scheme is 75% of that of the VTO scheme. In addition, compared with the STO scheme, the TTO scheme not only takes less time and energy, but also requires no sliding distance. Therefore, according to the simulation results of optimal trajectory planning for the above three schemes, the TTO scheme is the optimal scheme, in line with the time efficiency and energy consumption requirements for the BWTR.

The flight quality of the UAV is also affected by interference factors, flight control laws, and control distribution strategies. Thus, as topics for future research, first, the optimal control assignment strategy should be investigated for the redundant actuators of novel tilt-rotor UAVs. Second, the optimal controller design should be investigated by using an appropriate control method for the aerodynamic interference factor. The controller will minimize the impact of interference factors on UAV flight quality. Finally, field flight tests should be implemented to enable novel tilt-rotor UAVs to achieve optimal flight quality based on the proposed optimal takeoff scheme and trajectory, a reasonable control distribution strategy, and a robust anti-jamming controller.

## Figures and Tables

**Figure 1 sensors-22-09736-f001:**
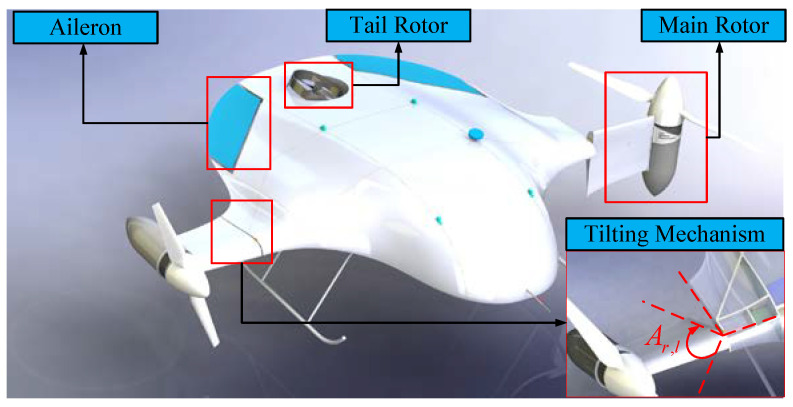
The structural platform of the BWTR.

**Figure 2 sensors-22-09736-f002:**
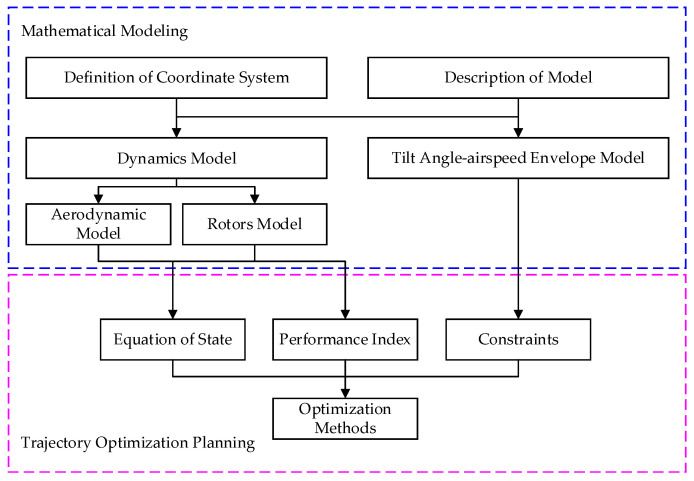
The structure of mathematical modeling framework.

**Figure 3 sensors-22-09736-f003:**
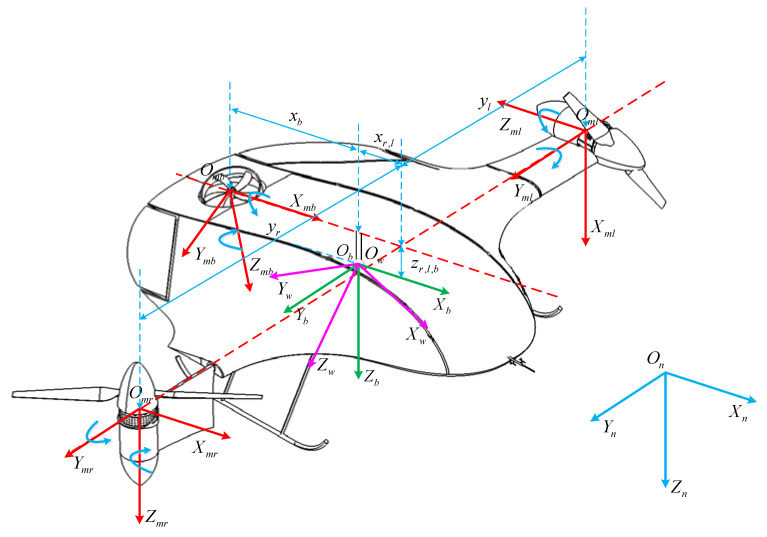
The structure of the BWTR UAV.

**Figure 4 sensors-22-09736-f004:**
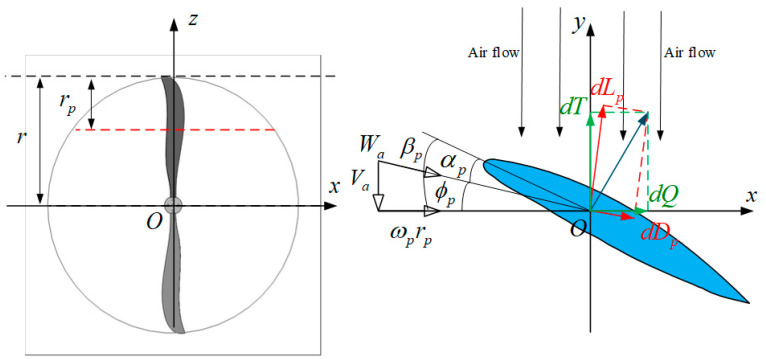
The force analysis diagram of the blade-element theory.

**Figure 5 sensors-22-09736-f005:**
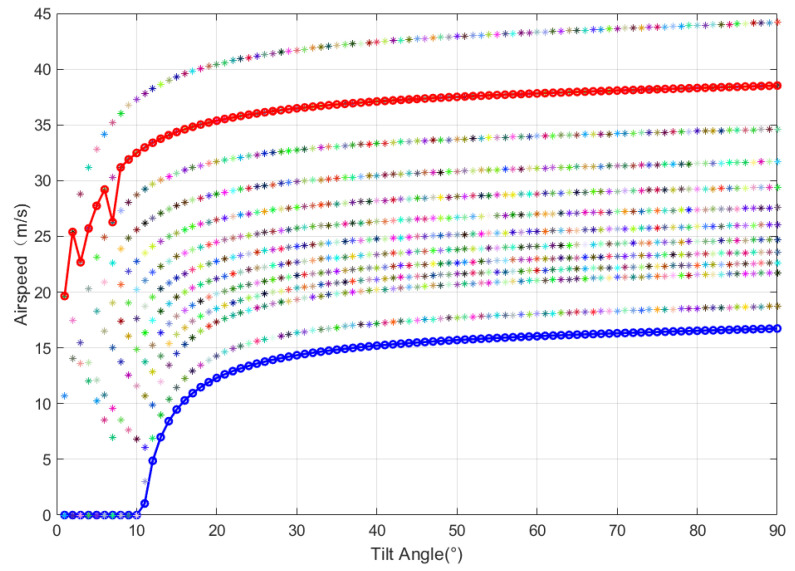
The boundary of the tilting corridor.

**Figure 6 sensors-22-09736-f006:**
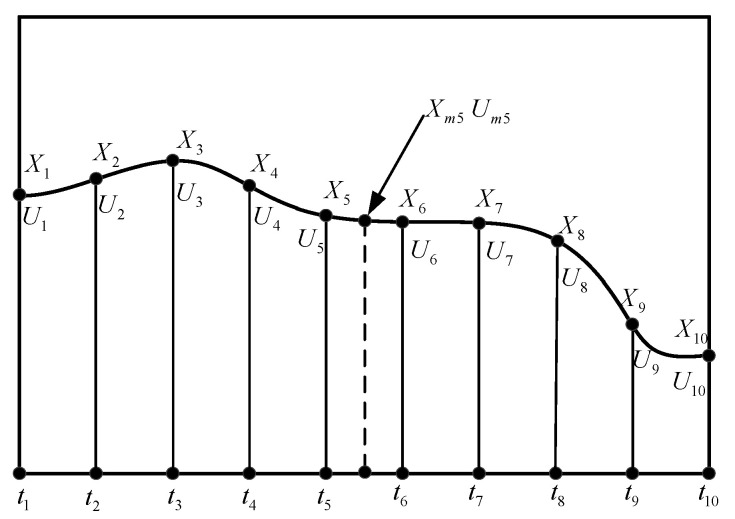
The direct collocation method.

**Figure 7 sensors-22-09736-f007:**
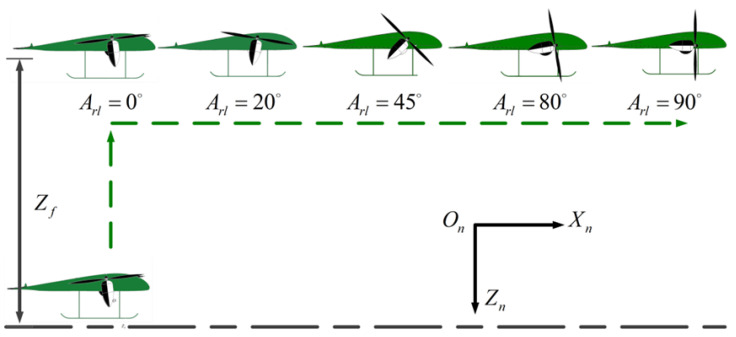
The flight path of VTO scheme.

**Figure 8 sensors-22-09736-f008:**
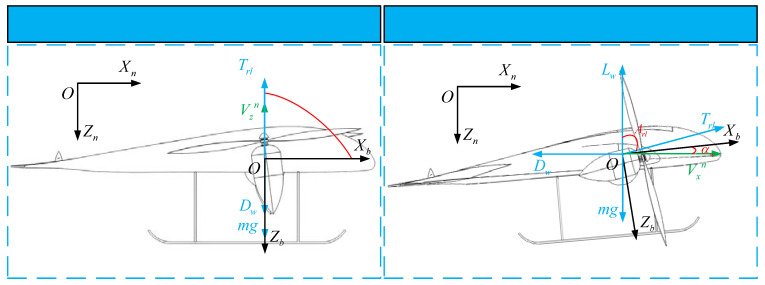
Force analysis in vertical and tilting stages of the VTO.

**Figure 9 sensors-22-09736-f009:**
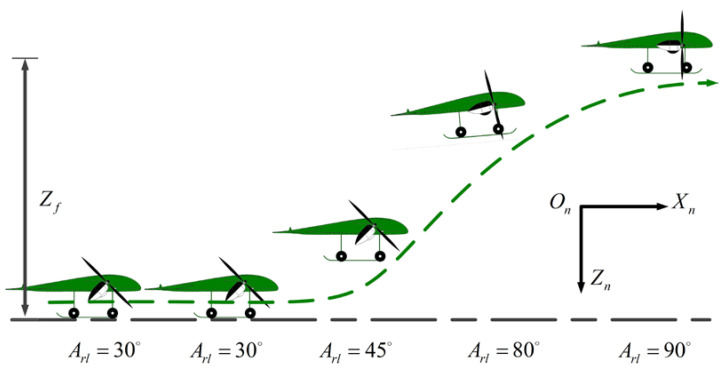
The flight path of STO scheme.

**Figure 10 sensors-22-09736-f010:**
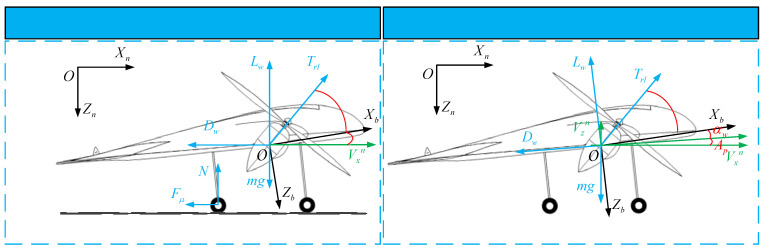
Force analysis in taxiing and takeoff stages of the STO.

**Figure 11 sensors-22-09736-f011:**
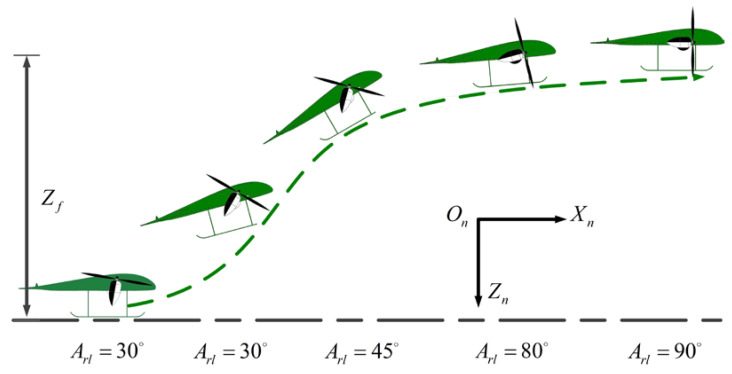
The flight path of TTO scheme.

**Figure 12 sensors-22-09736-f012:**
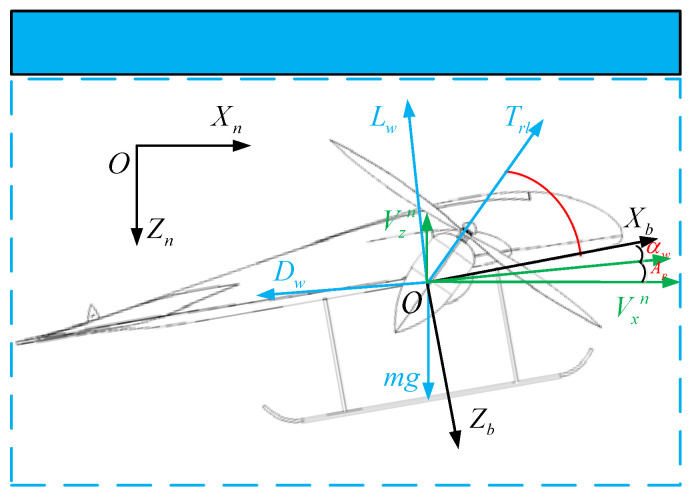
Forces analysis in TTO.

**Figure 13 sensors-22-09736-f013:**
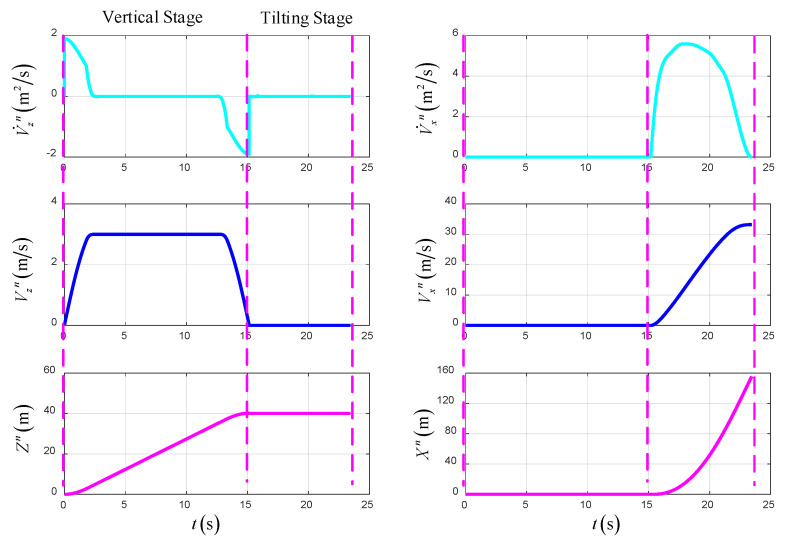
The state variables in VTO.

**Figure 14 sensors-22-09736-f014:**
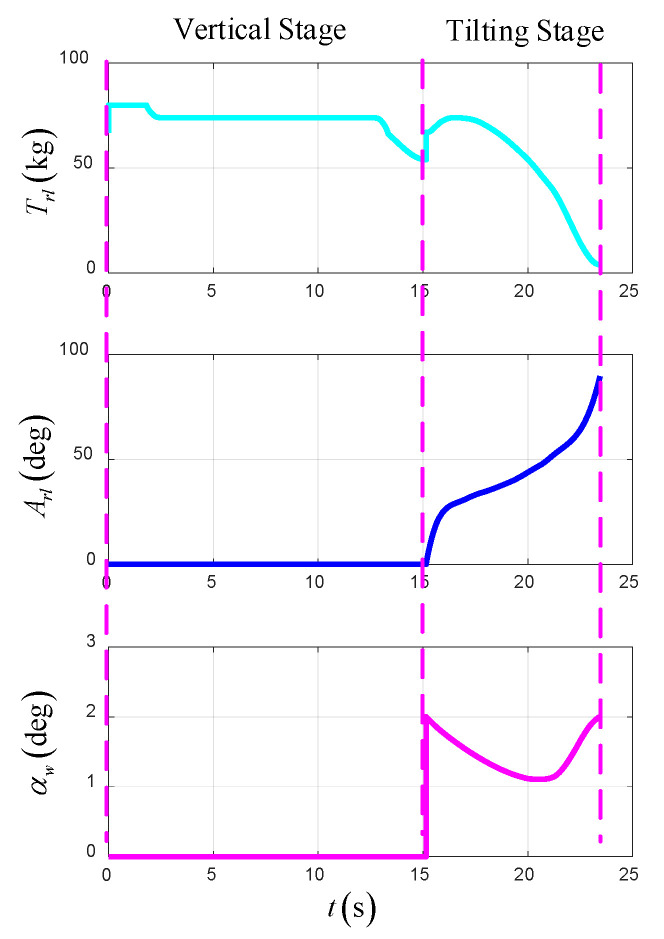
The control variables in VTO.

**Figure 15 sensors-22-09736-f015:**
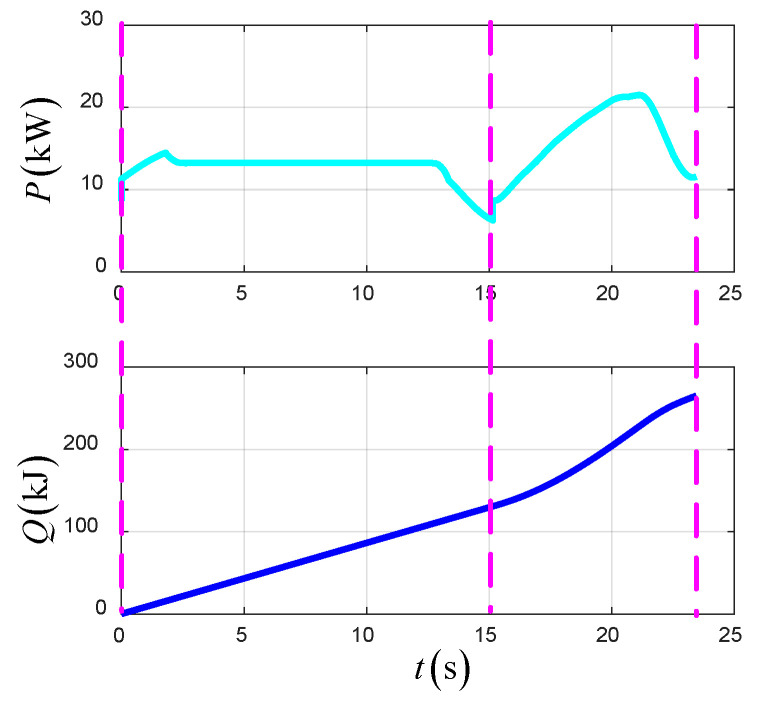
The power and energy consumption in VTO.

**Figure 16 sensors-22-09736-f016:**
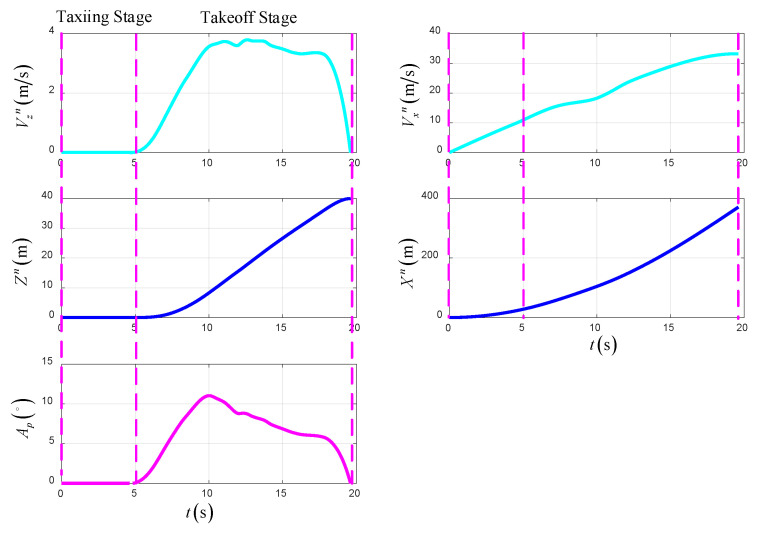
The state variables in STO.

**Figure 17 sensors-22-09736-f017:**
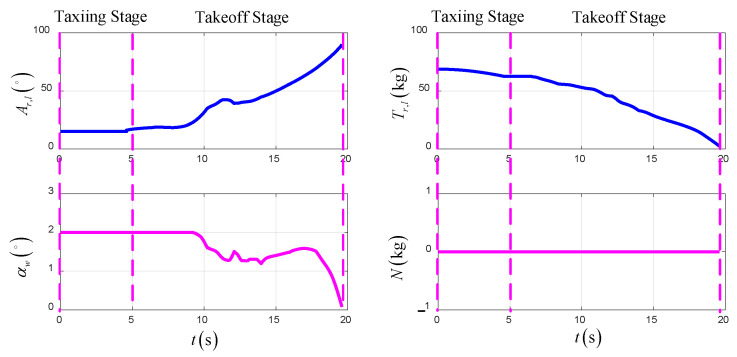
The control variables in STO.

**Figure 18 sensors-22-09736-f018:**
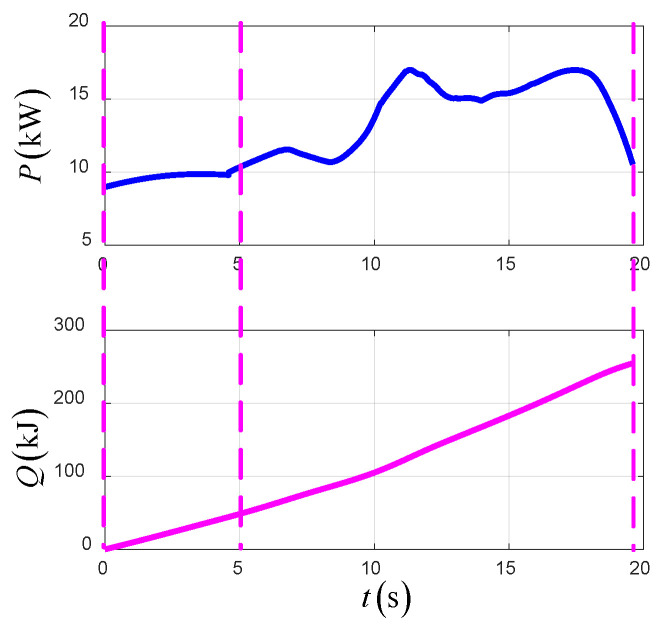
The power and energy consumption in STO.

**Figure 19 sensors-22-09736-f019:**
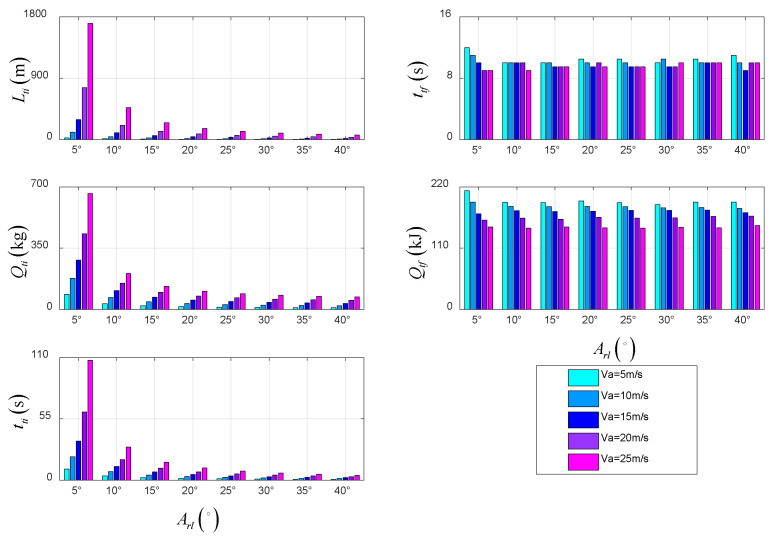
Performance parameters in taxiing and takeoff stage of STO.

**Figure 20 sensors-22-09736-f020:**
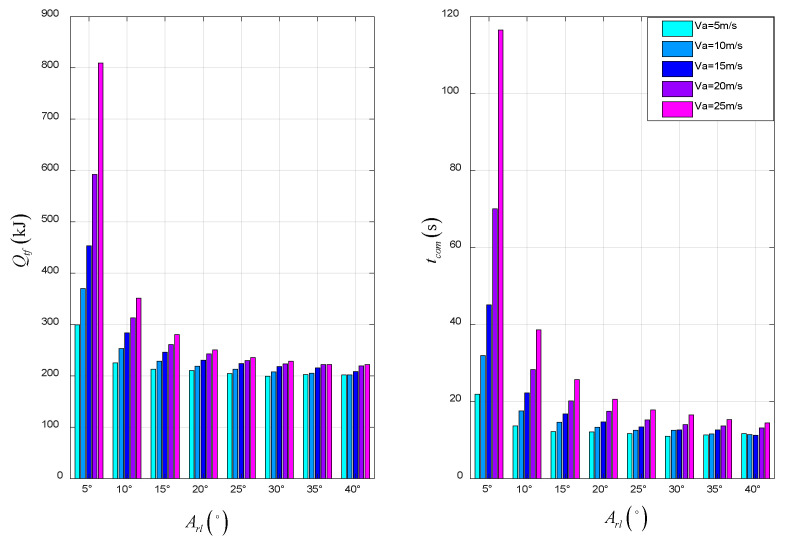
The total energy consumption and time in STO.

**Figure 21 sensors-22-09736-f021:**
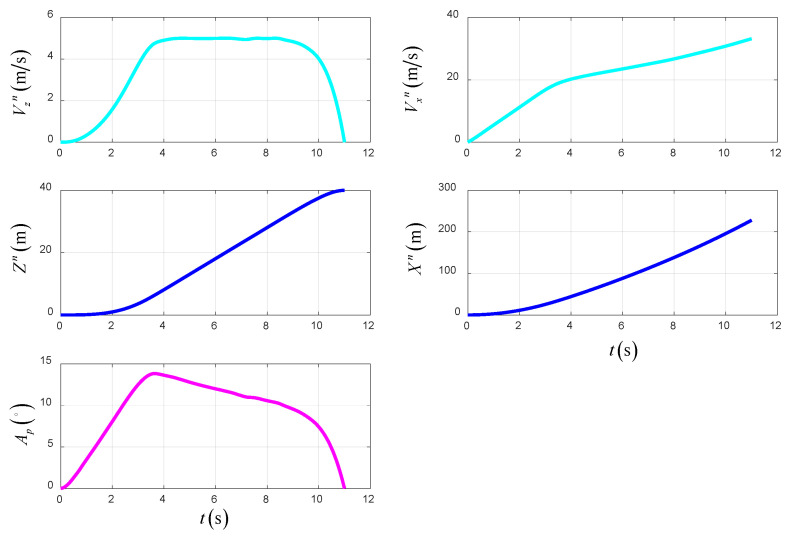
The state variables in TTO.

**Figure 22 sensors-22-09736-f022:**
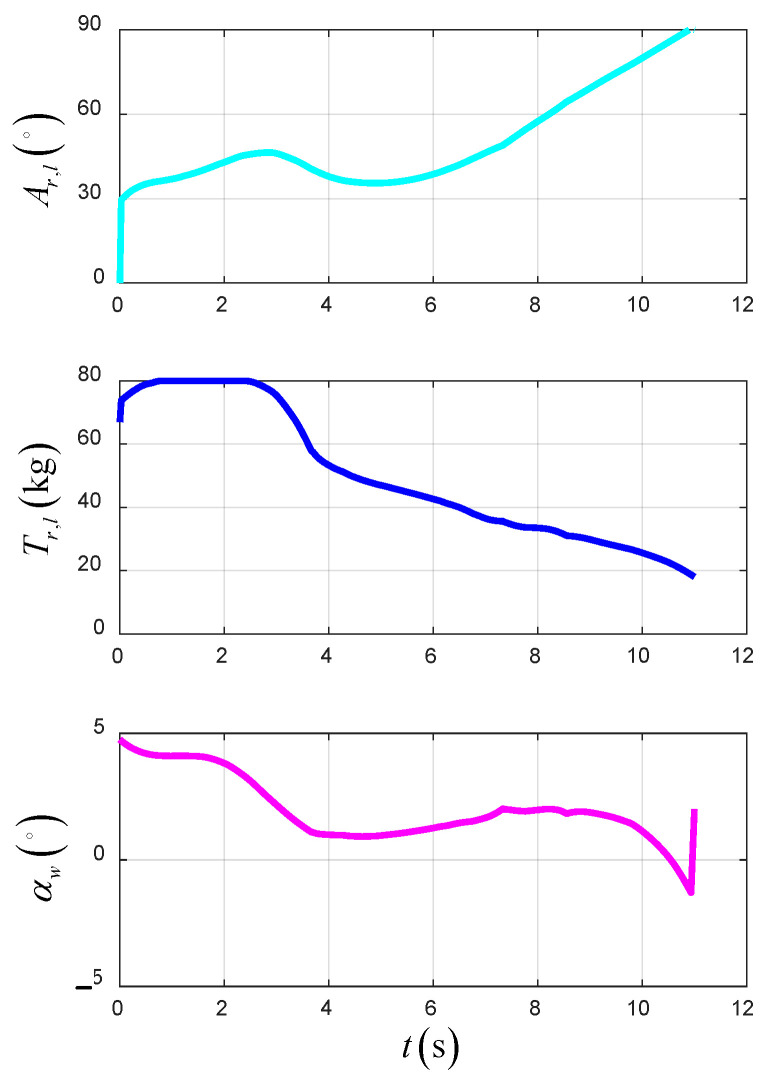
The control variables in TTO.

**Figure 23 sensors-22-09736-f023:**
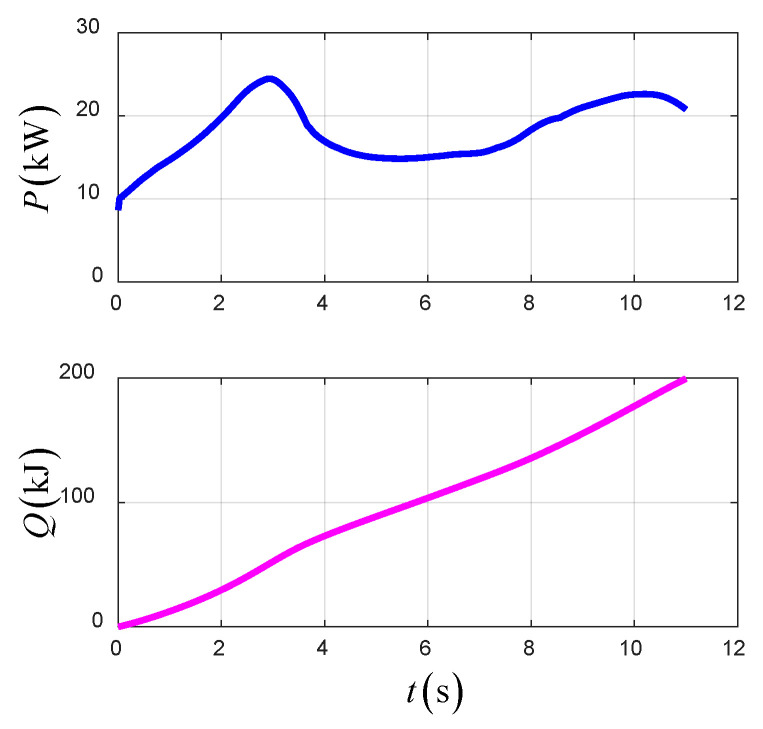
The power and energy consumption in TTO.

**Figure 24 sensors-22-09736-f024:**
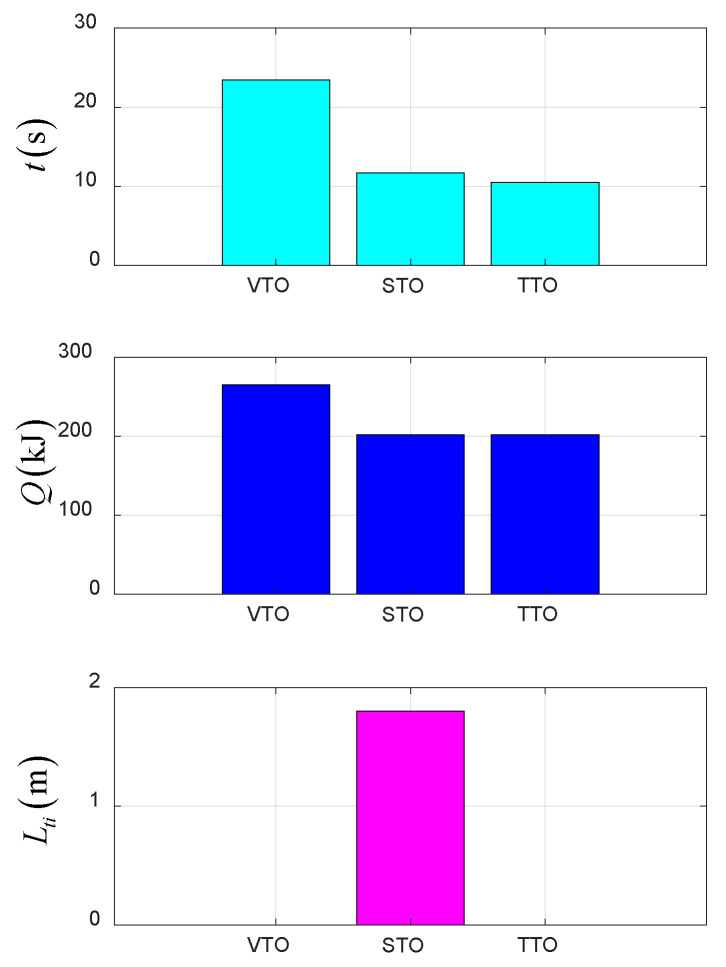
Comparison diagram of simulation results for three schemes.

**Table 1 sensors-22-09736-t001:** The model parameters of the tilt-rotor UAV.

Parameter	Definition	Value	Unit
m	Mass	70	kg
$I_{xx}$	Roll inertia	43.91	kg·m^2^
$I_{yy}$	Pitch inertia	15.13	kg·m^2^
$I_{zz}$	Yaw inertia	57.21	kg·m^2^
b	Wing span	3.15	m
$S_{w}$	Wing aera	4.01	m^2^
$c_{A}$	Chord length	1.27	m
${\left(x_{r},y_{r},z_{r}\right)}^{T}$	R rotor position	${\left(0.05,1.75,0.03\right)}^{T}$	m
${\left(x_{l},y_{l},z_{l}\right)}^{T}$	L rotor position	${\left(0.05,-1.75,0.03\right)}^{T}$	m
${\left(x_{b},y_{b},z_{b}\right)}^{T}$	B rotor position	${\left(-0.85,0,0.08\right)}^{T}$	m

**Table 2 sensors-22-09736-t002:** Main parameters used in the rotor modeling.

Parameter	Definition	Value	Unit
$H_{p}$	Pitch of propeller	0.30	m
r	Radius of propeller	0.46	m
$r_{P}$	Radius of the blade element	0.28	m
$c_{P}$	Average geometric chord length	0.05	m
λ	Correction factor of propeller	0.51	
$c_{fd}$	Zero-lift drag coefficient	0.02	
$B_{P}$	Number of blades	2	

**Table 3 sensors-22-09736-t003:** UAV variables in STO.

$A_{rl}\;\left({}^{\circ}\right)$	$V_{a}\;\left(m/s\right)$	$L_{ti}\;\left(m\right)$	$t_{ti}\;\left(s\right)$	$Q_{ti}\;\left(\mathrm{kJ}\right)$	$t_{tf}\;\left(s\right)$	$Q_{tf}\;\left(\mathrm{kJ}\right)$	$Q_{com}\;\left(\mathrm{kJ}\right)$	$t_{com}\;\left(s\right)$
5	5	25.2	9.9	85.9	12	213.2	299.1	21.9
	10	109.2	20.9	177.4	11	192.5	369.9	31.9
	15	290.6	35.1	281.7	10	171.6	453.3	45.1
	20	759.2	61.1	431.9	9	160.6	592.5	70.1
	25	1705	107.5	661.2	9	148.3	809.5	116.5
10	5	9.3	3.7	33.01	10	192.4	225.4	13.7
	10	39.2	7.6	68.5	10	184.8	253.3	17.6
	15	97.4	12.2	106.7	10	177.3	284	22.2
	20	206.2	18.3	149.4	10	163.8	313.2	28.3
	25	466.8	29.6	205.4	9	146.2	351.6	38.6
15	5	5.7	2.2	21.02	10	191.9	212.9	12.2
	10	23.7	4.6	44.4	10	184.3	228.7	14.6
	15	58	7.3	70	9.5	175.8	245.8	16.8
	20	119.1	10.7	98.5	9.5	162.2	260.7	20.2
	25	243.5	16.2	132.2	9.5	148.1	280.3	25.7
20	5	4	1.6	15.8	10.5	194.7	210.5	12.1
	10	16.7	3.3	33.9	10	184.8	218.7	13.3
	15	40.8	5.2	54.3	9.5	176.2	230.5	14.7
	20	82.8	7.5	77.3	10	165.7	243	17.5
	25	163.8	11.1	104.1	9.5	146.4	250.5	20.6
25	5	3.1	1.2	13	10.5	191.5	204.5	11.7
	10	12.8	2.5	28.2	10	184.4	212.6	12.5
	15	31.1	3.9	45.8	9.5	177.9	223.8	13.4
	20	62.6	5.7	66	9.5	163.9	229.9	15.2
	25	121.7	8.3	89.6	9.5	145.9	235.5	17.8
30	5	2.5	1	11.2	10	188.1	199.3	11
	10	10.2	2	24.7	10.5	182.7	207.4	12.5
	15	24.7	3.1	40.6	9.5	177.6	218.2	12.6
	20	49.6	4.5	59.1	9.5	164.4	223.5	14
	25	95.4	6.5	81	10	147.7	228.7	16.5
35	5	2	0.8	10	10.5	192.6	202.6	11.3
	10	8.4	1.6	22.5	10	182.8	205.3	11.6
	15	20.2	2.6	37.3	10	178.3	215.6	12.6
	20	40.34	3.693	54.73	10	167.1	222	13.7
	25	77.1	5.3	75.6	10	146.4	222	15.3
40	5	1.8	0.7	9.1	11	192.6	201.7	11.7
	10	7.1	1.4	20.7	10	181.3	202	11.4
	15	16.9	2.2	34.8	9	173.5	208.3	11.2
	20	33.5	3.1	51.6	10	167.7	219.3	13.1
	25	63.6	4.4	71.8	10	150.5	222.3	14.4

## Data Availability

Not applicable.

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
