# Peer review of "Study of Modeling and Optimal Take-Off Scheme for a Novel Tilt-Rotor UAV"

_sensors, 2022, doi:10.3390/s22249736_

Round 1

Reviewer 1 Report

See atttached

Author Response

Dear Editors and Reviewers,

Thank you very much for your careful review and constructive suggestions with regard to our manuscript " Research of Modeling and Optimal Take-off Scheme for a Novel Tilt-rotor UAV" (Manuscript ID: sensors-2022208). These comments are helpful for us to revise and improve our manuscript. We have studied comments carefully and tried our best to revise and improve the manuscript. Please see responses to the review comments in the accompanying file.

Reviewer 2 Report

The paper deals with the modeling and optimal Take-off scheme for a novel Tilt-rotor UAV. I have some observation to make:

The measurement unit writing in the Table 1 must be corrected.

The assumption ("The change in the position of the center of gravity of the UAV caused by the tilting of the motor can be ignored") made for the dynamic model should be explained. 

The paper is not easily readable. The Mathematical model section should be better explained in its organization, which should follow a clear and logical thread of steps. The sequence of steps taken should be explained upstream with their motivation.

It should also be explained how all the formulas that are presented are implemented and used in the next steps.

Each assumption must be explained in detail.

The originality, innovativeness and usefulness of the work presented should be more clearly highlighted and explained even starting from a more in-depth state of the art.

The presented results have been only carried out by simulations.

Author Response

(The authors gave the same response as above.)

Round 2

Reviewer 2 Report

The Authors have answered quite sufficiently to my observations. I suggest  to include the future developments of the research and to better justify the choice of presenting results only in simulation. The figure 5 should be checked.

Author Response

(The authors gave the same response as above.)
